# Genetic Links between Reproductive Traits and Amino Acid Pairwise Distances of Swine Leukocyte Antigen Alleles among Mating Partners in Microminipigs

**DOI:** 10.3390/ijms25137362

**Published:** 2024-07-04

**Authors:** Asako Ando, Tatsuya Matsubara, Shingo Suzuki, Noriaki Imaeda, Masaki Takasu, Atsuko Shigenari, Asuka Miyamoto, Shino Ohshima, Yoshie Kametani, Takashi Shiina, Jerzy K. Kulski, Hitoshi Kitagawa

**Affiliations:** 1Department of Molecular Life Science, Division of Basic Medical Science and Molecular Medicine, Tokai University School of Medicine, Isehara 259-1193, Japan; suzuki.shingo.t@tokai.ac.jp (S.S.); ashige@tokai.ac.jp (A.S.); vegetab1echi1dren@yahoo.co.jp (A.M.); shino-w@tokai.ac.jp (S.O.); y-kametn@tokai.ac.jp (Y.K.); tshiina@tokai.ac.jp (T.S.); or yurek.kulski@uwa.edu.au (J.K.K.); 2Joint Department of Veterinary Medicine, Faculty of Applied Biological Sciences, Gifu University, Gifu 501-1193, Japan; matsubara.tatsuya.b8@f.gifu-u.ac.jp (T.M.); noriaki-imaeda@ma.ctk.ne.jp (N.I.); 3Gifu University Institute for Advanced Study, Gifu University, Gifu 501-1193, Japan; takasu.masaki.i4@f.gifu-u.ac.jp; 4Center for One Medicine Innovative Translational Research (COMIT), Gifu University, Gifu 501-1193, Japan; 5Faculty of Health and Medical Sciences, School of Biomedical Science, The University of Western Australia, Crawley, WA 6009, Australia; 6Laboratory of Veterinary Internal Medicine, Faculty of Veterinary Medicine, Okayama University of Science, 1-3 Ikoino-oka, Imabari, Ehime 794-8555, Japan

**Keywords:** amino acid distance, litter sizes, microminipig, reproductive trait, swine leukocyte antigen

## Abstract

Previously, we found that a greater dissimilarity in swine leukocyte antigen (*SLA*) class I and class II alleles between mating partners resulted in increased farrowing rates in a highly inbred population of Microminipigs (MMPs). In this follow-up study, we have analyzed the effects of dissimilarity in *SLA* alleles between mating partners for seven different reproductive traits, including litter size and the number of stillborn and live or dead weaned piglets. We determined the relationships among reproductive traits within each mating event and the amino acid distances of *SLA* alleles as markers of diversity between mating partners. Our results indicate that mating partners with greater amino acid pairwise genetic distances in the *SLA-1* class I gene or *DQB1* class II gene alleles were associated with significantly larger litter sizes and higher numbers of live piglets at birth and weaning. Also, partners with greater pairwise distances in the *SLA-2* class I gene alleles exhibited fewer pre-weaning deaths. These findings suggest that the dissimilarity in *SLA* class I and class II alleles between mating partners may affect not only farrowing rates but also other key reproductive traits such as litter size and improved piglet survival rates. Consequently, *SLA* alleles could serve as valuable genetic markers for selecting mating partners in breeding programs and for conducting epistatic studies on various reproductive traits in MMPs.

## 1. Introduction

Pigs are essential large-animal models for biomedical research focused on reproductive functions, including maternal–fetal interactions, embryo development, and sperm studies [1,2]. The Microminipig (MMP) was developed by Fuji Micra Inc. as a novel miniature pig with an extremely small body size suitable for use in laboratory biomedical research [3]. The body measurements (weight, height, chest width, and chest circumference) of MMPs at 4 to 6 months of age were significantly smaller compared to those of young adult beagle dogs at 10 months old [3,4,5]. In the population of MMPs, eleven swine leukocyte antigen (SLA) class I and II haplotypes, including three recombinant haplotypes, were identified in the 14 parents or progenitors of the highly inbred MMP herd. These haplotypes consisted of a total of 25 class I alleles (nine alleles of the *SLA-1*, eight alleles of the *SLA-2*, and eight alleles of the *SLA-3* genes) and 14 class II alleles (eight alleles of the *DRB1* and six alleles of the *DQB1* genes). Homozygous MMPs with a high resolution haplotype (Hp-) 35.0 had two different alleles at the *SLA-1* gene, *SLA-1^*^12:01* and *SLA-1^*^13:01*, suggesting duplicated *SLA-1* genes [6]. 

The involvement of *SLA* genes in various reproductive functions has been observed with associations between specific *SLA* haplotypes and several important reproductive traits. These SLA associations include effects on the male genital tract development, ovulation rate and farrowing rate in females, litter size, stillbirth, preweaning mortality, and piglet weight at birth and weaning [1,4,7,8,9,10,11,12,13]. In addition, the *SLA* complex seems to play a role in piglet mortality in a swine line selected for resistance to mycoplasmal pneumonia [14]. 

In humans, certain human leukocyte antigens (HLA) shared between couples in closely related populations have been linked to infertility and miscarriages [15]. Furthermore, studies in humans and bovines indicate that sharing certain maternal–fetal or paternal MHC antigens can affect fetal development and survival [15,16]. Similarly, in MMPs, we recently reported that mating partners with shared *SLA* class II haplotypes–specifically, *DRB1* and/or *DQB1* alleles–tend to have lower farrowing rates compared to those with non-shared haplotypes. Moreover, lower farrowing rates were observed in mating pairs with smaller amino acid pairwise genetic distances among *SLA-1*, *SLA-3*, *DRB1*, and *DQB1* alleles. Also, a greater dissimilarity in *SLA* class II haplotypes or alleles between partners seems to be more beneficial for farrowing rates than dissimilarity among *SLA* class I haplotypes or alleles [17]. These findings align with the general hypothesis that many animal species, including rodents [18,19,20], humans [21,22], non-human primates [22], horses [23], pandas [24], birds [25], and fish [26], tend to choose mates with greater MHC dissimilarity. Therefore, *MHC* heterozygosity appears to enhance reproductive success by providing greater immune diversity to increase the chances of a successful pregnancy and the birth of healthy offspring.

In this study, we assessed the impact of *SLA* class I and class II genes on seven reproductive traits in MMPs, i.e., litter size at birth, number of live piglets in the litter size at birth, number of stillborn piglets in the litter size at birth, number of weaned piglets, number of suckling (pre-weaning) dead piglets, survival rates at birth, and weaning rates. We specifically examined the genetic effects of amino acid pairwise distances among *SLA-1*, *SLA-2*, *SLA-3*, *DRB1*, and *DQB1* alleles between mating pairs on these seven reproductive traits. This analysis aimed to better understand how SLA gene variability influences these other reproductive outcomes rather than just focusing on farrowing rates.

## 2. Results

### 2.1. Frequencies of SLA Class I and II Genotypes and Haplotypes

Table 1 shows that the highest SLA class I allele frequency (39.81%) was observed in sows or boars with *SLA-1^*^12:01*, *SLA-1^*^13:01*, *SLA-3^*^05:02*, or *SLA-2^*^10:01*. The second highest *SLA* class I allele frequency (24.12%) was observed in sows or boars with *SLA-1^*^11:04*, *SLA-3^*^04:01*, or *SLA-2^*^04:02:01*. The lowest SLA class I allele frequency (0.35%) was observed in sows or boars with *SLA-1^*^08:05*, *SLA-3^*^06:01*, or *SLA-2^*^05:04* (Table 1). These *SLA* class I alleles with the highest, second highest, and lowest frequencies were present within three SLA class I haplotypes (Hps), Hp-35.0, Hp-43.0, and Hp-6.0, respectively (Table 2). The remaining intermediate SLA class I allele frequencies, 6.96%, 6.54%, 10.81%, 7.81%, and 3.62%, representing various other SLA class I alleles in the three class I genes, were present within the five other class I Hps, Hp-10.0, -16.0, -17.0, -20.0, and -31.0, respectively (Table 2). The percentages of SLA class I and class II homozygous individuals in 104 sows and 42 boars were 12.5% and 21.4%, respectively.

With respect to frequencies of SLA class II alleles in mating sows and boars, the highest *DRB1* allele frequency (34.19%) was *DRB1^*^10:01*, and the highest *DQB1* allele frequency (41.08%) was *DQB1^*^06:01* (Table 1). The second highest *SLA* class I allele frequency (24.12%) was *SLA-1^*^11:04*. Table 1 shows that nine types of class I-*SLA-1* alleles and eight types of class I-*SLA-2* and *SLA-3* alleles and class II-*DRB1* alleles were found in the MMP population. The allele frequency of *DQB1^*^06:01*, 41.08% (Table 1), was calculated on the basis of its presence in three Hps, 0.7, 0.16, and 0.23. Because *DQB1^*^06:01* was shared with class II-Hps, 0.7, 0.16, or 0.23, only six types of SLA class II-*DQB1* alleles were found in the mating sows and boars (Table 2). In addition, the allele frequencies for the *SLA* class II-*DRB1* and -*DQB1* genes with recombinant Hps, Hp-35.17, Hp-43.17, or Hp-10.23 (corresponding to class II-Hps 0.17, 0.23, and 0.37), differ from the frequencies of the SLA class I-*SLA-1*, *SLA-2*, and *SLA-3* genes with class I-Hps 17.0, 35.0, and 43.0, respectively.

### 2.2. Reproductive Performances

Table 3 shows the results of reproductive performances after 650 mating events involving 148 MMPs. The mean values of litter sizes at birth and the number of weaned piglets were 5.47 ± 0.08 (SE) and 4.06 ± 0.09 (SE) piglets/delivery, respectively. These mean values of litter sizes and weaned piglets were similar to those previously reported in 2019: 5.48 ± 0.04 (SE) and 3.88 ± 0.04 (SE) piglets/delivery, respectively, using 2288 matings of 187 MMPs [13]. The mean value of the number of stillborn piglets at birth in 650 mating events (0.93 piglets/delivery in Table 3) was also similar to the results of 2288 matings (0.97 piglets/delivery) of 187 MMPs that were previously reported [13]. The survival rate at birth was 82.9%, and the weaning rate was slightly higher at 85.0% (Table 3).

### 2.3. Effects of Amino Acid Pairwise Distances between SLA Class I Alleles of Mating Pairs on Reproductive Traits in MMPs

The amino acid distances among nine *SLA-1*, eight *SLA-3*, and eight *SLA-2* alleles for all the mating pairs showed various values of substitutions per site at the class I loci ranging between 0.0480 to 0.2932, 0.0188 to 0.2184, and 0.1340 to 0.3206, respectively. The overall mean distances among the *SLA-1*, *SLA-3*, and *SLA-2* alleles were 0.1972, 0.1178, and 0.2330, respectively (Table 4). 

Table 5A,B show that the maximum sum values of the amino acid pairwise distances among four alleles in the *SLA-1* (*SLA-1^*^12:01* and *SLA-1^*^13:01)*, *SLA-3*, and *SLA-2* genes between the mating partners were 0.910, 0.994, 0.797, and 1.154, respectively. In these *SLA* loci, the sum of the pairwise distances were divided into nine to twelve range groups across 0.1 increments of substitutions per site. MMPs with Hp-35.0 have duplicated *SLA-1* genes that encode two alleles, *SLA-1^*^12:01* and *SLA-1^*^13:01* (Table 2). Therefore, the sum of amino acid pairwise distances of *SLA-1* alleles of mating pairs was calculated separately for *SLA-1^*^12:01* and *SLA-1^*^13:01* (Table 5A,B). The sum of the amino acid pairwise distances among the four alleles for each of the *SLA* class I genes carried by each mating pair was classified into nine to eleven range groups, which increased in distance from each other by increments of 0.1.

Figure 1A shows that the litter sizes at birth increased with a high positive correlation (n = 11, rs = 0.809, *p* = 0.003) in accordance with an increase in amino acid pairwise genetic distances of *SLA-1* alleles. Table 6 shows that significant positive correlations were observed between the litter sizes at birth (n = 11, rs = 0.809, *p* = 0.003) or weaning (n = 11, rs = 0.745, *p* = 0.008) and *SLA-1* alleles when using the pairwise distances with *SLA-1^*^13:01*. Additionally, a significant positive correlation (n = 9, rs = 0.817, *p* = 0.007) was observed between the litter size at birth and amino acid distances of *SLA-3* alleles for the mating partners. Significant positive correlations (n = 10, rs = 0.733, *p* = 0.016 or n = 11, rs = 0.773, *p* = 0.005) also were observed between the number of live piglets at birth and amino acid distances of *SLA-1* alleles using the pairwise distances with *SLA-1^*^12:01* or *SLA-1^*^13:01*, respectively, for mating partners. A slightly weaker, but significant positive correlation (n = 11, rs = 0.638, *p* = 0.035) was detected between the weaning rate and amino acid distances of *SLA-2* alleles of mating partners. In contrast, higher numbers of pre-weaning dead piglets were observed in mating pairs with relatively smaller pairwise genetic distances of *SLA-2* alleles. Also, a significant negative correlation (n = 11, rs = −0.752, *p* = 0.008) was observed between the number of pre-weaning dead piglets and amino acid pairwise distances of *SLA-2* alleles of mating partners (Table 6 and Figure 1B). No significant correlations were observed between three other types of reproductive traits, i.e., the number of stillborn piglets, survival rate at birth and weaning rate, and the *SLA* class I alleles of mating partners (Table 6).

### 2.4. Effects of Amino Acid Pairwise Distances between SLA Class II Alleles of Mating Pairs on Reproductive Traits in MMPs

Amino acid distances among eight *DRB1* alleles of mating pairs ranged between 0.0705 and 0.2287 substitutions per site, and the overall mean distance among the *DRB1* alleles was 0.1695. In contrast to the *DRB1* alleles in this and a previous study [17], the amino acid distances among six *DQB1* alleles of mating pairs were substantially lower, ranging between 0.0057 and 0.0867, with an overall mean distance of 0.0651 substitutions per site (Table 4).

Table 5A,B show the sum of the amino acid pairwise distances in each of the 650 matings at birth and 617 matings at weaning among four *DRB1* alleles that were classified into nine groups that accumulated from 0 to 0.899 across 0.1 incremental ranges. The grouped ranges of the sum of the pairwise distances from 0 to 0.831 among the *DRB1* alleles between mating partners were comparable with those of the class I-*SLA-1* and *SLA-3* alleles (Table 5A,B). In contrast to the nine or eight different *SLA-1*, *SLA-2*, *SLA-3*, and *DRB1* alleles, the amino acid distances among six *DQB1* alleles of mating pairs showed much lower values ranging between 0.0057 and 0.0867, with an overall mean distance of 0.0651 (Table 4). Moreover, a significant negative correlation (n = 9, rs = −0.678, *p* = 0.045) was observed between the number of pre-weaning dead piglets and amino acid pairwise distances of *DRB1* alleles among the mating partners. No significant correlation (*p* > 0.05) was detected among the six other reproductive traits, except the number of pre-weaning dead piglets and the amino acid pairwise distances of *DRB1* alleles of the mating partners (Table 6). 

Due to the narrow distribution of the amino acid pairwise distances and the small number of different *DQB1* alleles among mating partners, the sum of the amino acid pairwise distances in each of the 650 matings at birth and 617 matings at weaning was classified into only eight range groups that accumulated from 0 to 0.311 across 0.05 increments (Table 5A,B). However, Figure 1C and Table 6 show a significant positive correlation (n = 8, rs = 0.976, *p* < 0.001) between the litter sizes at birth or weaning and the amino acid pairwise distances of *DQB1* alleles. A significant negative correlation (n = 8, rs = −0.714, *p* = 0.047) or positive correlation (n = 8, rs = 0.738, *p* = 0.037) was observed also among the number of pre-weaning dead piglets or weaning rates, respectively, and the amino acid pairwise distances of *DQB1* alleles of the mating partners. Furthermore, significant positive correlations (n = 8, rs = 0.952, *p* < 0.001) were obtained between the number of live piglets at birth and amino acid distances of the mating partners’ *DQB1* alleles.

## 3. Discussion

Litter sizes have been analyzed as a measure of reproductive performance in the pig populations of various breeds of domestic pigs and several minipigs, such as Göttingen and NIBS minipigs [27,28,29,30]. In this study, we confirmed that the mean values of litter sizes and weaned piglets in MMPs were slightly larger than those in NIBS minipigs: 4.4 ± 1.5 (SD) and 3.4 ± 1.3 (SD), respectively [27], even though the body sizes of the MMPs were considerably smaller than the NIBS minipigs produced in Japan [3,5]. The mean values of litter sizes and number of weaned piglets in MMPs were also similar to those of other laboratory-used miniature pigs, such as the Göttingen and other breeds of domestic pigs [9,28,29]. Furthermore, the 85% weaning rate of MMPs was only slightly lower than the mean value of those of 112 commercial pig farms in England [30]. Thus, the MMPs population used in the present study showed relatively normal reproductive traits that are useful for evaluating the relationships among the reproductive traits and amino acid distances of *SLA* alleles between mating pairs. 

We found in our previous study that dissimilarities in *SLA* class I and class II alleles between mating pairs had significantly influenced farrowing rates in MMPs, and higher farrowing rates were observed in mating pairs with greater amino acid pairwise genetic distances for the *SLA-1* (using *SLA-1^*^13:01)*, *SLA-3*, *DRB1*, and *DQB1* alleles [17]. We extended this analysis into the present study by examining whether the same *SLA* gene alleles and haplotypes in different mating partners affected other reproductive traits, such as litter sizes and number of stillbirths and sucklings. We discovered that mating pairs with larger amino acid pairwise genetic distances for the *SLA-1*, *SLA-3*, or *DQB1* alleles also tended to have more favorable reproductive outcomes, including larger litter sizes and more live piglets at birth and weaning. However, we also found that mating pairs with smaller amino acid pairwise genetic distances for the *SLA-2* alleles exhibited unfavorable reproductive traits, such as a higher rate of pre-weaning dead piglets.

The observed influence of MHC-based mating preferences tends to be relatively weak or variable in various mammalian species, including rodents [19,20], humans [21,31], and non-human primates [22]. For example, in primates, the Fisher’s Z correlation coefficient for dissimilarity was Zr = 0.044 [22]. Furthermore, studies in humans have demonstrated that MHC dissimilarity affects mate choice marginally across various loci, including MHC classes I and II [32]. However, in this study, as well as in our previous report on farrowing rates [17], we found significant effects of SLA dissimilarity between partners on reproductive traits in pigs, specifically at the *SLA-1*, *SLA-2*, *SLA-3*, and *DQB1* loci. These findings suggest that *SLA* class I and class II allele dissimilarities between mating partners might contribute to better reproductive traits, such as larger litter sizes, a greater number of live piglets at birth, more piglets at weaning, and lower pre-weaning mortality rates, in the MMP population.

The exact mechanisms responsible for the variable effects on reproductive traits caused by amino acid dissimilarity between partners are not yet fully understood. Regarding SLA class I and II expression during pregnancy, cell-type-specific regulation of the *SLA* and *B2M* genes in uterine cells is influenced by progesterone and interferons (IFNs) [33]. However, the correlation between the success of fertilization, depending on SLA sharing, and the stage-specific expression of SLA class I molecules during pregnancy remains unclear. It is known that SLA class I molecules are not expressed in the pig placenta during pregnancy. However, SLA class II-DQ molecules increase in response to conceptus-derived interferon gamma (IFNG) during pregnancy, suggesting a role in regulating the immune response to maintain pregnancy at the maternal–fetal interface [34]. This report on the expression of SLA-DQ molecules in uterine endometrial tissues implies that these SLA molecules might affect embryonic growth rates and litter size. 

In our study, despite a limited range of pairwise distances and a small number of different *DQB1* alleles, we observed strong significant correlations (*p* < 0.001) among the litter sizes, the number of live piglets at birth, and the number of weaned piglets with amino acid pairwise distances of *DQB1* alleles between mating pairs. These correlations had lower *p*-values than those among the three reproductive traits and the pairwise distances of *SLA-1* (**13:01*) or *SLA-3*. The trends for stronger correlations among the three reproductive traits and the pairwise distances of *DQB1* alleles align with our previous results [17], which showed a high positive correlation between farrowing rates and amino acid pairwise distances of *DQB1* alleles between mating pairs, as compared to other *SLA* class I or class II alleles. These results suggest that dissimilarities of *DQB1* alleles between mating pairs may have a more crucial role on reproductive performance than those of other *SLA* class I or class II alleles. Furthermore, maternal MHC has specific effects, i.e., uterine natural killer (uNK) cells, educated by maternal MHC, do not reject fetuses that lack self-MHC, allowing these fetuses to grow to their full potential [35]. 

According to Koketsu et al. [36], several critical factors influence the reproductive performance of female breeding pigs. These factors include both sow-level and herd-level variables, such as breeding temperature, parity, birth weight, lactation feed intake, lactation length, insemination timing, and age at first mating. These factors are crucial for enhancing reproductive performance. On the other hand, dissimilarity in *SLA* class I and II alleles between mating pairs appears to have relatively weak and inconsistent effects on four reproductive traits, i.e., litter size, number of live piglets at birth, the number of weaned piglets, and pre-weaning mortality, as well as farrowing rates. This suggests that the *SLA* class I and class II genes might be indirectly responsible for improved reproductive performance by way of secondary immunity mechanisms and factors. Despite these limitations, the findings of our study indicate that *SLA* class I and class II alleles or haplotypes could be valuable genetic markers for selecting mating pairs in breeding programs. This could lead to more effective breeding management of MMPs, even though the correlation between SLA dissimilarity and reproductive performance might be indirect or auxiliary. 

Genome-wide association studies (GWAS) recently have investigated reproductive traits, particularly litter size and the number of live piglets at birth. Numerous single nucleotide polymorphisms (SNPs) and various candidate genes associated with increased litter size or a higher number of live piglets at birth were mapped to specific chromosomal regions in Large White, Landrace, Duroc, and Yorkshire pig lines [37,38,39,40]. Bekoev et al. reported in their review [40] that SNPs found in the Pig Quantitative Trait Loci database (PigQTLdb) associated with reproductive traits are often located in genes, with about 60% found within introns. However, there is still no consensus on the optimal approach to conducting GWAS due to the wide range of statistical methodologies available for such studies. Given these uncertainties, further research is needed to understand how these candidate genes identified by GWAS [40] and the *SLA* genes that we have identified might interact epistatically and influence the reproductive traits in MMPs. 

Because our current study on the effects of SLA dissimilarity between mating partners on reproductive performance used only a limited number of *SLA* class I and II gene alleles and haplotypes in a small population and a single breed, further research is required to validate the impact of SLA dissimilarity on pregnancy and successful deliveries in other pig breeds with different *SLA* gene alleles and haplotypes.

## 4. Materials and Methods

### 4.1. Animals

MMPs used in this study were bred as a herd at Fuji Micra Inc. (Fujinomiya, Shizuoka, Japan) from June 2008 to February 2017. The reproductive outcomes obtained for a total of 650 cumulative mating events involving 106 sows and 42 boars as mating pairs, which resulted in 3557 piglets, were used for our present genetic analyses. These sows and boars were clinically healthy and did not show any clinical manifestation. The maximum body weight was 26.0 kg and 23.3 kg in a sow at two years and eleven months of age and a boar at four years and six months of age, respectively. The breeding sows and boars were assigned to eleven different *SLA* class I and class II haplotypes, including three recombinant ones. The matings of MMPs were random. However, during some generations, especially during initial matings, mating pairs with relatively small body sizes were preferentially selected to establish the characteristics of the MMP breed [3].

This study was approved by the Animal Care and Use Committee of Gifu University (#17042, 26 May 2017). The care and use of the laboratory animals were conducted in compliance with the guidelines of Good Laboratory Practice of Gifu University and Fuji Micra Inc., Shizuoka, Japan

### 4.2. SLA Class I and Class II Typing

Polymorphic SLA alleles for three class I (*SLA-1*, *SLA-2*, and *SLA-3*) and two class II (*DRB1* and *DQB1*) genes were assigned by low-resolution *SLA* genotyping in 148 breeding MMPs (106 sows and 42 boars) using a PCR-sequence-specific primers (SSP) method, as described previously [6]. These SLA alleles were used to deduce eight specific types of high-resolution SLA class I and class II haplotypes, Hp-6.7, Hp-10.11, Hp-16.16, Hp-17.17, Hp-20.18, Hp-31.13, Hp-35.23, and Hp-43.37, and three class I and class II recombinant haplotypes, Hp-10.23, Hp-35.17, and Hp-43.17. These haplotypes were deduced from a preliminary analysis of the inheritance and segregation of alleles of the three class I (*SLA-1*, *SLA-2*, and *SLA-3*) and two class II genes (*DRB1* and *DQB1*), respectively, in descendants of the MMP population as described previously [6] (Table 2). To analyze the association between *SLA* class I and class II alleles and seven reproductive traits in this study (Section 4.3), the *SLA* alleles in four-digit genotypes of the 148 breeding MMPs were deduced from the two-digit genotypes for the three class I (*SLA-1*, *SLA-2*, and *SLA-3*), and two class II (*DRB1* and *DQB1*) genes and the eleven types of low-resolution *SLA* class I and class II haplotypes (Table 2).

### 4.3. Measurements of Reproductive Performances

The seven reproductive traits analyzed were litter size, number of live and stillborn piglets, number of weaned piglets, number of pre-weaning dead piglets, survival rate, and weaning rate. Litter sizes were measured at birth, including the total number of live and stillbirth piglets in 650 deliveries. The total number of live and stillborn piglets at birth was recorded separately by adding together the number of live births, stillbirths, and newborn piglets, respectively. The numbers of surviving weaned piglets and pre-weaning dead piglets were measured separately as a weaning number of live piglets and a number of piglets that died during suckling (“pre-weaning”). Survival rates at birth were calculated as the ratios of live newborn piglets per litter size. Weaning rates were calculated as the ratios of surviving weaned piglets per live newborn piglets.

### 4.4. Influence of Amino Acid Distance of SLA Class I and Class II Genotypes between Mating Partners on Reproductive Performances

To evaluate the relationship between the seven reproductive traits and amino acid divergences of SLA class I (*SLA-1*, *SLA-2*, and *SLA-3*) or class II (*DRB1* and *DQB1*) alleles in each mating pair, amino acid pairwise distances among the alleles were analyzed by MEGA X software using the JTT matrix-based model with a gamma distribution as described previously [17]. The pairwise distances were calculated based on the number of amino acid substitutions per site between their allele sequences [41,42]. The pairwise distances among *SLA-1*, *SLA-2*, *SLA-3*, *DRB1*, or *DQB1* alleles that have been already published in our previous report were used in this study [16]. The range and mean values of the pairwise distances among *SLA-1*, *SLA-2*, *SLA-3*, *DRB1*, or *DQB1* alleles are summarized in Table 4. The sum of all the pairwise amino-acid distances (D) of the four possible alleles (A, a, B, and b) in each mating pair was calculated as follows:Alleles with sire: A and a.Alleles with dam: B and b.The sum of all the pairwise amino-acid distances = (DAB + DAb + DaB + Dab).If the mating pair has identical amino acid sequences (same alleles), the sum of all the pairwise amino acid distances is ‘0’.

The sum of the pairwise amino acid distances (DAB + DAb + DaB + Dab) of *SLA* alleles in each *SLA* class I or class II locus between mating pairs was divided into nine to twelve range groups across 0.1 increments. The number of matings in each of the range groups was calculated for each *SLA* locus (Table 5A,B). 

The sum of the pairwise amino acid distances among four alleles in each of the *SLA* class I genes and class II-*DRB1* gene carried by each mating pair was classified into nine to eleven range groups, which increased in distance from each other by 0.1 increments. On the other hand, the sum of the amino acid pairwise distances among the *SLA* four alleles of the class II-*DQB1* gene carried by each mating pair was classified into eight range groups that increased in distance from each other by 0.05 increments (Table 5A,B). Correlation coefficients were calculated between the central value of amino acid pairwise distance and mean or rate of reproductive traits in each range group.

### 4.5. Statistical Analyses

Correlations between seven reproductive traits (litter size at birth, number of live piglets at birth, number of stillborn piglets at birth, number of weaned piglets, number of pre-weaning dead piglets, survival rates at birth or weaning rates) in each range group and the central value of amino acid pairwise distances of each SLA locus between mating partners were evaluated by Spearman’s rank correlation coefficient (BellCurve in Excel, Social Survey Research Information Co., Ltd. Tokyo, Japan); *p*-values of less than 0.05 were considered significant. We used this non-parametric statistical test instead of the parametric Pearson’s correlation for all correlations because, according to the Shapiro–Wilk normality test, some of our data sets were not distributed normally.

## 5. Conclusions

The dissimilarities of SLA class I and class II alleles between mating partners appear to be involved in various reproductive traits such as litter sizes, number of live piglets at birth, number of weaned piglets, and number of pre-weaning dead piglets. These relationships suggest that the amino acid distance of SLA alleles between the mating pairs is related not only to farrowing rate but also to other productivity traits of pigs. These relationships among the reproductive traits and SLA show that *SLA* alleles are potentially useful genetic markers for selection of mating partners in breeding programs and epistatic studies of various reproductive traits of MMPs.

## Figures and Tables

**Figure 1 ijms-25-07362-f001:**
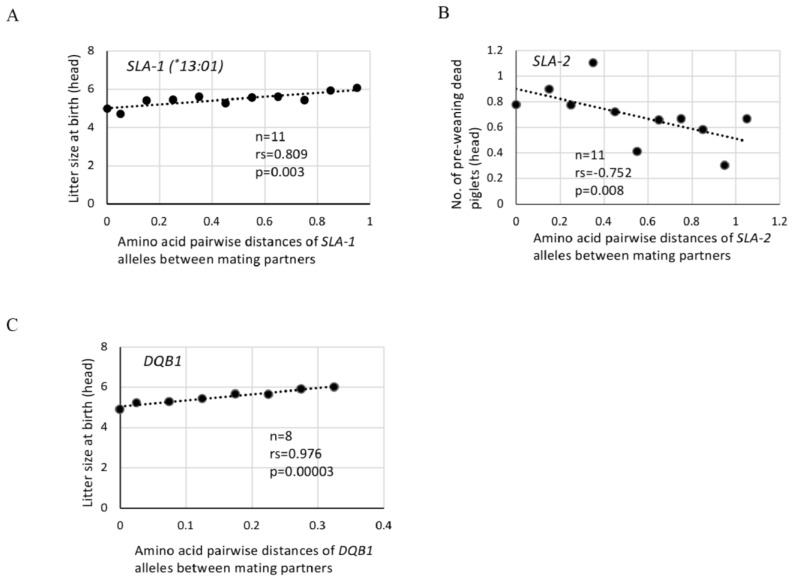
Effects of amino acid pairwise distances of *SLA* class I and class II alleles between partners on litter size at birth and number of piglets that died during lactation in MMPs. (**A**,**C**): Dots represent the litter size at birth (head) using the center value of each group of amino acid pairwise distances of *SLA-1* (*^*^13:01*) (**A**) and *DQB1* (**C**) alleles of mating partners as shown in Table 5A. (**B**): Dots represent the number of dead piglets during lactation (head) using the center value of each group of amino acid pairwise distances of *SLA-2* (**B**) alleles of mating partners as shown in Table 5B. rs: rs (ρ) is Spearman’s rank correlation coefficient, and p: p value evaluated by Spearman’s rank correlation coefficient.

**Table 1 ijms-25-07362-t001:** Allele frequencies of the *SLA*-class I and II genes in MMPs.

	SLA Class I allele						SLA Class II allele			
No.	*SLA-1*	(%)	*SLA-3*	(%)	*SLA-2*	(%)	*DRB1*	(%)	*DQB1*	(%)
1	*^*^08:05*	0.35	*^*^06:01*	0.35	*^*^03:02*	6.96	*^*^09:01*	6.96	*^*^06:01*	41.08 ^a^
2	*^*^05:01*	6.96	*^*^08:01*	6.96	*^*^03:02*	6.96	*^*^09:01*	6.96	*^*^04:02*	6.96
3	*^*^04:01*	6.54	*^*^06:02*	6.54	*^*^09:01*	6.54	*^*^11:03*	6.54	*^*^05:01*	17.92 ^b^
4	*^*^08:04*	10.81	*^*^03:05*	10.81	*^*^06:03*	10.81	*^*^08:01*	17.92 ^b^	*^*^04:01:02*	7.81
5	*^*^10:02*	7.81	*^*^01:01*	7.81	*^*^11:01:02*	7.81	*^*^14:01*	7.81	*^*^03:03*	3.62
6	*^*^15:02*	3.62	*^*^07:01:02*	3.62	*^*^16:01*	3.62	*^*^04:03*	3.62	*^*^05:02*	22.62 ^b^
7	*^*^12:01, ^*^13:01*	39.81	*^*^05:02*	39.81	*^*^10:01*	39.81	*^*^10:01*	34.19 ^b^		
8	*^*^11:04*	24.12	*^*^04:01*	24.12	*^*^04:02:02*	24.12	*^*^07:01*	22.62 ^b^		

^a^ *DQB1^*^06:01* is present in Hp-0.7, 0.16, and 0.23. Therefore, the allele frequency of *DQB1^*^06:01* is calculated as the total numbers in MMPs with these three Hps. ^b^ For the three recombinant Hps, Hp-35.17, 43.17, or 10.23, the allele frequencies of the *SLA* class II*-DRB1* and *DQB1* genes in MMPs with class II-Hps, 0.17, 0.23, and 0.37 are different from the allele frequencies of the *SLA* class I-*SLA-1*, *SLA-2*, and *SLA-3* genes in MMPs with class I-Hps, 17.0, 35.0, and 43.0. (%) indicates the frequency of each SLA class I or class II allele.

**Table 2 ijms-25-07362-t002:** *SLA*-class I and II genotypes and number of *SLA* haplotypes in MMPs.

No.	Haplo-Type (Hp-)	*SLA*-Class I			*SLA*-Class II		Number of Haplotypes ^c^			
							(Frequency (%))			
		*SLA-1* ^a^	*SLA-3* ^a^	*SLA-2* ^a^	*DRB1* ^a^	*DQB1* ^a^	Dams		Sires	
1	6.7	*^*^08:05*	*^*^06:01*	*^*^05:04*	*^*^06:01*	*^*^06:01*	9	(0.69)	0	(0)
2	10.11	*^*^05:01*	*^*^08:01*	*^*^03:02*	*^*^09:01*	*^*^04:02*	102	(7.85)	79	(6.08)
3	16.16	*^*^04:01*	*^*^06:02*	*^*^09:01*	*^*^11:03*	*^*^06:01*	83	(6.38)	87	(6.69)
4	17.17	*^*^08:04*	*^*^03:05*	*^*^06:03*	*^*^08:01*	*^*^05:01*	179	(13.77)	102	(7.85)
5	20.18	*^*^10:02*	*^*^01:01*	*^*^11:01:02*	*^*^14:01*	*^*^04:01:02*	144	(11.08)	59	(4.54)
6	31.13	*^*^15:02*	*^*^07:01:02*	*^*^16:01*	*^*^04:03*	*^*^03:03*	63	(4.85)	31	(2.38)
7	35.23	*^*^12:01,^*^13:01*	*^*^05:02*	*^*^10:01*	*^*^10:01*	*^*^06:01*	353	(27.15)	536	(41.23)
8	43.37	*^*^11:04*	*^*^04:01*	*^*^04:02:02*	*^*^07:01*	*^*^05:02*	300	(23.08)	288	(22.15)
9	10.23 ^b^	*^*^05:01*	*^*^08:01*	*^*^03:02*	*^*^10:01*	*^*^06:01*	0	(0)	0	(0)
10	35.17 ^b^	*^*^12:01,^*^13:01*	*^*^05:02*	*^*^10:01*	*^*^08:01*	*^*^05:01*	35	(2.69)	111	(8.54)
11	43.17 ^b^	*^*^11:04*	*^*^04:01*	*^*^04:02:02*	*^*^08:01*	*^*^05:01*	32	(2.46)	7	(0.54)

^a^ Allele specificity assigned by low-resolution typing at the two-digit level represents the allele specificity detected by high-resolution typing in Microminipigs. *DQB1^*^04:02* (Hp-0.11) and *DQB1^*^04:01:02* (Hp-0.18), and *DQB1^*^05:01* (Hp-0.17) and *DQB1^*^05:02* (Hp-0.37) are assigned as *DQB1^*^04:XX* (Lr-0.11 or Lr-0.18) and *DQB1^*^05:XX* (Lr-0.17 or Lr-0.37) using a PCR-SSP method, respectively. ^b^ Hp-10.23, Hp-35.17, and Hp-43.17 were assigned as recombinant haplotypes [6]. ^c^ The numbers of haplotypes in dams and sires are indicated as the sum of the number of diploid haplotypes that are calculated in 650 matings using 106 dams and 42 sires as mating partners.

**Table 3 ijms-25-07362-t003:** Summary of reproductive traits in 148 MMPs.

Developmental Stage	Trait	No. of Matings ^a^	Mean	Standard Error
At birth	Mean litter size	650	5.47	0.08
	Mean No. of live piglets	650	4.54	0.09
	Mean No. of stillborn piglets	650	0.93	0.06
	Survival rate	650	82.9 ^b^	
At weaning	Mean No. of weaned piglets	617	4.06	0.09
	Mean No. of pre-weaning dead piglets	617	0.68	0.05
	Weaning rate	617	85.0 ^b^	

^a^ The data availability for the number of matings and reproductive traits was 650 at birth (litter size, number of live piglets, number of stillborn piglets, and rate of live piglet), but only 617 at weaning (number of weaned piglets, number of pre-weaning dead piglets, and weaning rate), respectively. ^b^ Survival rate and weaning rate are presented as percentages (%).

**Table 4 ijms-25-07362-t004:** Amino acid pairwise distances among *SLA-1*, *SLA-3*, *SLA-2*, *DRB1*, or *DQB1* alleles.

Gene	No. of Alleles	Amino Acid Pairwise Distance	
		Range	Mean ^a^
*SLA-1*	9	0.0480–0.2932	0.1972
*SLA-3*	8	0.0188–0.2184	0.1178
*SLA-2*	8	0.1340–0.3206	0.2330
*DRB1*	8	0.0705–0.2287	0.1695
*DQB1*	6	0.0057–0.0867	0.0651

^a^ Overall mean distance was calculated using amino acid pairwise distances among *SLA* alleles in each of the *SLA* class I and class II genes.

**Table 5 ijms-25-07362-t005:** (A) Number (No.) of matings with four reproductive traits at birth (litter size, No. of live piglets, No. of stillbirths, and survival rate) linked to a progressive and incremental range of the sum of the amino acid pairwise distances of *SLA* class I and class II alleles between mating pairs. (B) Number (No.) of matings with three reproductive weaning traits (litter size, No. of dead piglets during lactation, and weaning rate) linked to a progressive and incremental range of the sum of the amino acid pairwise distances of *SLA* class I and class II alleles between mating pairs.

**(A)**									
**Amino Acid Pairwise**		**No. of Matings ^a^**					**Amino Acid Pairwise**		**No. of**
**Distance**							**Distance**		**Matings**
**Range**	**Center**	** *SLA-1* **	** *SLA-1* **	** *SLA-3* **	** *SLA-2* **	** *DRB1* **	**Range**	**Center**	** *DQB1* **
	**Value**	**(*^*^12:01*)**	**(*^*^13:01*)**					**Value**	
0	0	10	10	10	10	6	0	0	18
0.01–0.099	0.05	0	24	29	0	124 ^b^	0.001–0.049	0.025	107
0.1–0.199	0.15	133	109	149	44		0.05–0.099	0.075	109
0.2–0.299	0.25	52	74	144	141	86	0.10–0.149	0.125	138
0.3–0.399	0.35	56	31	156	19	126	0.15–0.199	0.175	112
0.4–0.499	0.45	55	87	48	43	69	0.20–0.249	0.225	118
0.5–0.599	0.55	115	96	50	36	146	0.25–0.299	0.275	33
0.6–0.699	0.65	116	47	47	141	67	0.30–0.349	0.325	15
0.7–0.799	0.75	48	112	17	119	20			
0.8–0.899	0.85	56	36		56	6			
0.9–0.999	0.95	9	24		35				
1.0–1.099	1.05				6				
^a^ Total number of matings for four reproductive traits at birth (litter size, No. of live piglets, No. of stillbirths, and survival rate) are 650 as shown in Table 3. ^b^ Due to only one mating at a range group by the sum of the pairwise distance from 0.01 to 0.099 among *DRB1* alleles between partners, this range group and another one, 0.1–0.199, were combined as one range group, 0.01–0.199 (center value = 0.10).									
**(B)**									
**Amino Acid Pairwise**		**No. of Matings ^a^**					**Amino Acid Pairwise**		**No. of**
**Distance**							**Distance**		**Matings**
**Range**	**Center**	** *SLA-1* **	** *SLA-1* **	** *SLA-3* **	** *SLA-2* **	** *DRB1* **	**Range**	**Center**	** *DQB1* **
	**Value**	**(*^*^12:01*)**	**(*^*^13:01*)**					**Value**	
0	0	9	9	9	9	5	0	0	17
0.01–0.099	0.05	0	21	24	0	117 ^b^	0.001–0.049	0.025	102
0.1–0.199	0.15	124	103	141	39		0.05–0.099	0.075	101
0.2–0.299	0.25	48	68	135	133	83	0.10–0.149	0.125	132
0.3–0.399	0.35	54	30	149	19	120	0.15–0.199	0.175	110
0.4–0.499	0.45	52	84	48	43	67	0.20–0.249	0.225	109
0.5–0.599	0.55	109	91	49	34	135	0.25–0.299	0.275	31
0.6–0.699	0.65	110	46	45	132	67	0.30–0.349	0.325	15
0.7–0.799	0.75	48	107	17	114	18			
0.8–0.899	0.85	54	35		55	5			
0.9–0.999	0.95	9	23		33				
1.0–1.099	1.05				6				
^a^ Total number of matings for three reproductive traits at weaning (No. of weaned piglets, No. of pre-weaning dead piglets, and weaning rate) are 617, as shown in Table 3. ^b^ Due to only one mating at a range group by the sum of the pairwise distance from 0.01 to 0.099 among *DRB1* alleles between partners, this range group and another one, 0.1–0.199, were combined as one range group, 0.01–0.199 (center value = 0.10).									

**Table 6 ijms-25-07362-t006:** Effects of amino acid pairwise distances of SLA class I and class II alleles between partners on reproductive traits in MMPs.

	Gene		*SLA-1* (**12:01*)	*SLA-1* (**13:01*)	*SLA-3*	*SLA-2*	*DRB1*	*DQB1*
Developmental Stage	Trait	n	10	11	9	11	9	8
At birth	Litter size	rs	0.382	0.809	0.817	0.255	0.217	0.976
		*p*	0.276	0.003	0.007	0.450	0.576	<0.001
	No. of live piglets	rs	0.733	0.773	0.536	0.255	0.233	0.952
		*p*	0.016	0.005	0.137	0.450	0.546	<0.001
	No. of stillborn piglets	rs	−0.139	0.318	−0.083	−0.409	−0.483	−0.071
		*p*	0.701	0.340	0.831	0.212	0.187	0.867
	Survival rate (%)	rs	0.479	0.291	0.050	0.555	0.233	0.619
		*p*	0.162	0.385	0.898	0.077	0.546	0.102
At weaning	No. of weaned piglets	rs	0.491	0.745	0.050	0.445	0.333	0.976
		*p*	0.150	0.008	0.898	0.170	0.381	<0.001
	No. of pre-weaning dead piglets	rs	−0.340	−0.573	−0.583	−0.752	-0.678	-0.714
		*p*	0.336	0.066	0.099	0.008	0.045	0.047
	Weaning rate (%)	rs	0.406	0.336	0.317	0.638	0.617	0.738
		*p*	0.244	0.312	0.406	0.035	0.077	0.037

n: number of groups on the sum of pairwise amino acid differences of *SLA-1* (**12:01* and **13:01*), *SLA-3*, *SLA-2*, *DRB1*, and *DQB1* alleles between mating partners as shown in Table 5A,B. rs: rs (ρ) is Spearman’s rank correlation coefficient, and *p*: *p* value evaluated by Spearman’s rank correlation coefficient. Underlines indicate *p* < 0.01.

## Data Availability

The original contributions presented in the study are included in the article; further inquiries can be directed to the corresponding authors.

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
