# Peer review of "Genetic Links between Reproductive Traits and Amino Acid Pairwise Distances of Swine Leukocyte Antigen Alleles among Mating Partners in Microminipigs"

_ijms, 2024, doi:10.3390/ijms25137362_

Round 1
Reviewer 1 Report
Comments and Suggestions for Authors
Here, the authors analyzed a huge number of matings between inbred microminipigs to assess, whether genetic differences in SLAs (swine leucocyted antigens) haplotypes had an effect on breeding success in terms of farrowing rate, litter size, and survival of piglets. These parameters were then correlated with genetic analysis of the SLA haplotypes.
The analyses were done retrospectively on several mating attempts in a breeding station of this inbred pig line and are descriptive in first line, however, the pig, and particularly, inbred minipig lines are an important resouce for biomedical research. Therefore a detailled characterization is of outmost interest, e.g. for future breeding directions to avoid inbreeding depression events.
An important piece of info missing from the manuscript is the inbreeding grade of these animals, i.e. howmany generations of inbreeding, homozygosity of these animals, overall health, and maximum weight.
Comments on the Quality of English Language
none
Author Response
Responses to Reviewers’ comments
Reviewer 1
- Here, the authors analyzed a huge number of matings between inbred microminipigs to assess, whether genetic differences in SLAs (swine leucocyted antigens) haplotypes had an effect on breeding success in terms of farrowing rate, litter size, and survival of piglets. These parameters were then correlated with genetic analysis of the SLA haplotypes.
Response: Thank you for your kind summary on our present study using SLA defined Microminipigs.
- The analyses were done retrospectively on several mating attempts in a breeding station of this inbred pig line and are descriptive in first line, however, the pig, and particularly, inbred minipig lines are an important resouce for biomedical research. Therefore a detailled characterization is of outmost interest, e.g. for future breeding directions to avoid inbreeding depression events.
Response: Thank you for your thoughtful comments on inbred minipig lines. In eleven SLA haplotypes of the Microminipig population, there is no observed impossible breeding by any SLA haplotype combinations between partners. Therefore, as experimental pigs with known SLA haplotypes, it will be important to maintain the minipig population with a certain level of SLA-haplotype variation for application of various studies such as transplantation, disease susceptibility, and vaccine research.
- An important piece of info missing from the manuscript is the inbreeding grade of these animals, i.e. how many generations of inbreeding, homozygosity of these animals, overall health, and maximum weight.
Response: In the Microminipig population, it would be difficult to calculate the inbreeding grade for a complicated breeding program in Fuji Micra Inc. On homozygosity of Microminipigs in this study, we have added the following sentence on line 99 in Results, “The percentages of SLA class I and class II homozygous individuals in 104 sows and 42 boars were 12.5% and 21.4%, respectively.”
Regarding overall health, we have added a following sentence on line 374 in Materials and Methods. “These sows and boars were clinically healthy, and did not show any clinical manifestation. The maximum body weight was 26.0 kg and 23.3 kg in a sow at two years and eleven months of age and a boar at four years and six months of age, respectively.”

Reviewer 2 Report
Comments and Suggestions for Authors
Dear authors,
An interesting article, however, I have a major concern relating to the correlation tests used. Please see bellow.
Line 261 give the reference of your previous study
Line 349 remove the word “basically”
In Table 1 please remove the blank line between sentences in notes. Again in table 2,… check all the others.
Also, make sure tables are not broken between pages
While calculating correlations you have used the non-parametric Spearman's test. Why did you do this? Why didn’t you use the Pearson's test (parametric)? Did you test the normal distribution of the variables? Which test did you use? What were the results?
Also, should you conclude by the non-parametric test, this should be represented by (ρ) rho not an “r”,… “r” should be used with Pearson’s correlation.
Author Response
Responses to Reviewers’ comments
Reviewer 2
- Line 261 give the reference of your previous study
Response:
The last sentences on line 261: We found in our previous study that dissimilarities in SLA class I and class II alleles between mating pairs had significantly influenced farrowing rates in MMPs. Higher farrowing rates were observed in mating pairs with greater amino acid pairwise genetic distances for the SLA-1 (using SLA-1*13:01), SLA-3, DRB1 and DQB1 alleles [17].
According to the Reviewer’s comment, we have revised above sentences as follows,
Revised sentences on line 282: We found in our previous study that dissimilarities in SLA class I and class II alleles between mating pairs had significantly influenced farrowing rates in MMPs, and higher farrowing rates were observed in mating pairs with greater amino acid pairwise genetic distances for the SLA-1 (using SLA-1*13:01), SLA-3, DRB1 and DQB1 alleles [17].
- Line 349 remove the word “basically”
Response: According to the Reviewer’s suggestion, we have removed the word “basically” as follows,
A revised sentence on line 378: The matings of MMPs were random.
- In Table 1 please remove the blank line between sentences in notes. Again in table 2,… check all the others.
Response: According to the Reviewer’s comments, we have removed the blank in Table 1. Moreover, we have also checked all other tables.
- Also, make sure tables are not broken between pages 
Response: We have revised Table 2 and Table 5 that were broken between pages.

- While calculating correlations you have used the non-parametric Spearman's test. Why did you do this? Why didn’t you use the Pearson's test (parametric)? Did you test the normal distribution of the variables? Which test did you use? What were the results?
Response: For the analysis of the genetic association between mean or rate of seven reproductive traits and the sum of the pairwise amino acid distances among four alleles in each of the SLA class I and class II genes, we first evaluated all data sets for normal distribution using Shapiro-Wilk normality test. We found that the results for some data sets [SLA-1(*12:01), SLA-3, DRB1, and DQB1 SLA-1(*12:01), SLA-3, DRB1, and DQB1] were not normally distributed, possibly because of small data numbers used in the analysis of eight to eleven range groups across 0.1 or 0.05 increments. Therefore, for overall consistency, we used the non-parametric Spearman’s ranked test for all the data sets. Pearson’s correlation assumes that the data is normally distributed and that the relationship between the variables is linear. This makes it less suitable for data that do not meet these assumptions. Consequently, we chose the Spearman’s rank correlation as the non-parametric version over the parametric Pearson product-moment correlation because it is statistically more robust and reliable than the Pearson product-moment correlation for a small number of data points, and it does not require the assumption of normality. We have added the following sentence to the end of our paragraph 4.5 Statistical analyses in Material and Methods at line 451:
“We used this non-parametric statistical test instead of the parametric Pearson’s correlation for all correlations because according to the Shapiro-Wilk normality test some of our data sets were not distributed normally. “
- Also, should you conclude by the non-parametric test, this should be represented by (ρ) rho not an “r”,… “r” should be used with Pearson’s correlation.
Response: We have agreed with the Reviewer’s suggestion. However, “r” is confusable with “p” when we use the both in Table 6 and Figure 1. Since “r” is signified by “rs”, we have revised “r” to “rs” in Table 6, Figure 1, and text. Furthermore, we have also revised a footnote on “r” in Table 6 and Figure 1 as follows,
rs: rs (r) is Spearman’s rank correlation coefficient, and p: p value evaluated by Spearman’s rank correlation coefficient.
